

# Automated detection of atmospheric $NO_2$ plumes from satellite data: a tool to help infer anthropogenic combustion emissions

Douglas P. Finch[1,2], Paul I. Palmer[1,2], and Tiaran Zhang[3]

[1]National Centre for Earth Observation, University of Edinburgh, UK
[2]School of GeoSciences, University of Edinburgh, UK
[3]National Centre for Earth Observation, Kings College London, UK, now at Satellite Vu, UK

**Correspondence:** Douglas Finch (d.finch@ed.ac.uk)

**Abstract.** We use a convolutional neural network (CNN) to identify plumes of nitrogen dioxide ($NO_2$), a tracer of incomplete combustion, from $NO_2$ column data collected by the TROPOspheric Monitoring Instrument (TROPOMI). This approach allows us to exploit efficiently the growing volume of satellite data available to characterise Earth's climate. For the purposes of demonstration, we focus on data collected between July 2018 and June 2020. We train the deep learning model using six
thousand $28{\times}28$-pixel images of TROPOMI data (corresponding to $\simeq266{\times}133$ km$^2$) and find that the model can identify plumes with a success rate of more than 90%. Over our study period, we find over 310,000 individual $NO_2$ plumes of which $\simeq$9% are found over mainland China. We have attempted to remove the influence of open biomass burning using correlative high-resolution thermal infrared data from the Visible Infrared Imaging Radiometer Suite (VIIRS). We relate the remaining $NO_2$ plumes to large urban centres, oil and gas production, and major power plants. We find no correlation between $NO_2$ plumes
and the location of natural gas flaring. We also find persistent $NO_2$ plumes from regions where inventories do not currently include emissions. Using an established anthropogenic $CO_2$ emission inventory, we find that our $NO_2$ plume distribution captures 92% of total $CO_2$ emissions, with the remaining 8% mostly due to a large number of small sources <0.2 gC/m$^2$/day for which our $NO_2$ plume model is less sensitive. We argue the underlying CNN approach could form the basis of a Bayesian framework to estimate anthropogenic combustion emissions.

## 1  Introduction

The Paris Agreement (PA) is the current inter-government vehicle that describes a progressive reduction in greenhouse gas (GHG) emissions to mitigate dangerous climate change, described as a larger than two-degree Celsius increase in global mean temperature above pre-industrial values. Whether it will achieve its stated goals depends on commitments of its signatories to establish and more importantly realise stringent plans to reduce effectively national GHG emissions. The PA includes
two main activities: quinquennial Global Stocktakes (GST) and Nationally Determined Contributions (NDC) that describe pledged emission reductions during successive GSTs. Given the implications of non-compliance and the need to make large and rapid emission reductions, measurement, reporting and verification (MRV) systems are being developed that will help guide nations on the effectiveness of policies (Janssens-Maenhout et al., 2020). The main focus on these MRV systems are anthropogenic emissions of carbon dioxide ($CO_2$) and methane. One of the challenges faced by all these MRV systems is





separating the anthropogenic and natural components of $CO_2$ and methane fluxes. Here, we use a deep learning model to identify automatically satellite-observed plumes of nitrogen dioxide ($NO_2$), a proxy for incomplete combustion, to locate combustion hotspots, e.g. oil and gas industry, cities, and powerplants.

Burning of fossil fuels, representing emissions of 9-10 PgC/yr (Friedlingstein et al., 2020), has been shown unequivocally to impact Earth's climate via rising atmospheric levels of gases such as $CO_2$ and methane that can absorb and radiate infra-

red radiation. The distribution of these emissions is heterogeneous across the globe, disproportionately focused on cities, oil and gas extraction facilities, energy generation facilities, and flows of physical trade that rely heavily on shipping and road transportation (Poore and Nemecek, 2018). Compiled inventories, which rely on self-reporting, provide estimates on these emissions but rely on assumptions that can sometimes lead to inaccurate values. Cities are responsible for almost three quarters of the fossil fuel contribution to atmospheric $CO_2$ (Edenhofer et al., 2014), but questions remain about the veracity of reported

emissions (e.g., Gurney et al. (2021)) and the disproportionate role of a small number of super-emitters (e.g., Duren et al. (2019)). We know where most power plants are geographically located but new and large coal-fired power plants continue to be built and commissioned in countries such as China and India, potentially compromising their short-term climate ambitions within the PA. The rate of their construction often outpaces updates to inventory estimates. International shipping only represents a few percent of global $CO_2$ emissions but they appear to be going up (Organization, 2020). Given the importance of

establishing accurate national emission baselines from which to reduce emissions as part of the PA, it is essential we have a robust measurement-based approach to estimate emissions of $CO_2$ and methane to complement inventory estimates.

A growing body of work has been using satellite observations to study point sources of $CO_2$ (Bovensmann et al., 2010; Kort et al., 2012; Hakkarainen et al., 2016; Nassar et al., 2017; Broquet et al., 2018; Brunner et al., 2019; Kuhlmann et al., 2019; Zheng et al., 2019; Wang et al., 2019; Kuhlmann et al., 2020; Strandgren et al., 2020; Wang et al., 2020; Wu et al.,

2020; Yang et al., 2020; Ye et al., 2020; Zheng et al., 2020) and methane (Varon et al., 2019; de Gouw et al., 2020; Varon et al., 2021), taking advantage of global measurement coverage, subject to clear skies. Even with the 0.3% precision of $CO_2$ columns detected by the NASA Orbiting Carbon Observatory-2 instrument, dilution of point source emissions across a 3 $km^2$ grid box could potentially result in the directly overhead column being elevated but not elevate the measurements immediately downwind except under exceptional circumstances. Other studies have recognized this shortcoming and have taken advantage

of trace gases that are co-emitted with $CO_2$ and methane during the combustion process. For many industrial combustion processes, air provides the source of molecular oxygen necessary for the fuel to burn. While molecular nitrogen ($N_2$) in the air does not take part in the combustion reaction, the temperatures involved can thermally dissociate $N_2$ to facilitate the production of NO (and to a lesser extent $NO_2$). In the absence of widespread use of scrubbers that remove nitrogen oxides from combustion exhaust and with the subsequent influence of photochemistry that rapidly interconverts NO and $NO_2$, $NO_2$ is widely assumed

to be a robust proxy for combustion $CO_2$ (Reuter et al., 2019; Liu et al., 2020; Hakkarainen et al., 2021; Ialongo et al., 2021). The main advantage of using $NO_2$ as a tracer of combustion is its atmospheric e-folding lifetime, which ranges from hours to a day in the lower troposphere. Consequently, any major surface emissions will result in an observable plume close to the point of emission.



All of these studies represent case studies or a small number of case studies, reflecting the difficulty of locating $CO_2$ plumes

and coincident measurements of $NO_2$. This piecemeal approach is inconsistent with the vast volume of data being produced by the current generation of satellite instruments, in particular the TROPOspheric Monitoring Instrument (TROPOMI), and limits our ability to quantify the changing influence of $CO_2$ hotspots on the global carbon cycle. Here, we address this issue by using a deep learning algorithm to detect automatically $NO_2$ plumes. This work builds on earlier remote sensing image detection studies that use machine learning, e.g. Lary et al. (2016); Maxwell et al. (2018). As we show, the number of plumes found

in any one year is $O(10^5)$, allowing us to study more systematically how $NO_2$ can be used to study combustion emission of carbon.

In section 2 we discuss the TROPOMI $NO_2$ and thermal anomaly data that we use to identify anthropogenic plumes of $NO_2$. We also describe the deep learning method we use, including our approach to supervised learning, which underpins our ability to detect automatically $NO_2$ plumes. In section 3 we report the performance of our $NO_2$ plume detection method, and use the

ensemble of plumes to assess how well it detects $CO_2$ emissions described by an established inventory.We conclude the paper in section 4, including a discussion of next steps.

## 2    Data and Methods

We describe the TROPOMI retrieved data of $NO_2$ columns that we use to study combustion, and VIIRS biomass burning data we use to isolate the influence of fossil fuel combustion. We also describe the development of our deep learning model to detect

$NO_2$ plumes.

### 2.1    Satellite Data

**TROPOMI Column Observations of $NO_2$**

We use level 2 retrieved tropospheric column $NO_2$ data retrieved from the TROPOspheric Monitoring Instrument (TROPOMI), launched in 2017. We use two years of $NO_2$ column data from July 2018 to June 2020. These data are taken from the Sentinel-

5P Pre-Operations Data Hub (https://s5phub.copernicus.eu/dhus/). For further information about these level 2 data products we refer the reader to studies dedicated to $NO_2$ (Boersma et al., 2010; Van Geffen et al., 2015; Lorente et al., 2017; Zara et al., 2018).

TROPOMI is a UV-Vis-NIR-SWIR spectrometer aboard the Copernicus Sentinel-5 Precursor (S5-P) satellite, which is in a Sun-synchronous orbit with a local equatorial overpass time of 13:30. TROPOMI has a swath width of 2600 km divided

into 450 across-track pixels for which during our study period have dimensions of 7 km×3.5 km (across×along track) for $NO_2$. This sampling strategy results in near-daily global coverage (Veefkind et al., 2012), subject to cloud-free scenes. In this study, we only use pixels with a quality flag >0.75, as recommended by the TROPOMI Level 2 Product User Manuals.Higher resolution data are available from August 2019, but for consistency we have used the original resolution throughout our study period.



**VIIRS Thermal Anomaly Data**

We use thermal anomaly data from the Visible Infrared Imaging Radiometer Suite (VIIRS) on board the Suomi National Polar-orbiting Partnership (NPP) satellite, launched in 2011 as a proxy to identify $NO_2$ plumes from biomass burning. We use the 375 meter Level-2 VNP14 product from https://firms.modaps.eosdis.nasa.gov/download/. VIIRS provides near twice-daily global coverage at a spatial resolution of 750 m. During the study period we found 16,056,612 vegetation fires spotted by VIIRS, after discarding low confidence data.

We attribute an $NO_2$ plume to biomass burning if it is within 15 km of a biomass burning scene identified by VIIRS. We chose that distance criterion because it corresponds to approximately two TROPOMI pixels and should account for any offset error in determining the plume centre. We find that a 5–10 km adjustment to this criterion does not significantly affect our results. Development of a more sophisticated method, taking account of other trace gas measurements, is outside the scope of this study.

For the purposes of this study, we discard biomass burning scenes to focus on anthropogenic combustion source but we acknowledge that the converse to this approach is also scientifically valid.

## 2.2 Deep Learning Model to Identify $NO_2$ Plumes

To automatically detect plumes of $NO_2$ from TROPOMI data, we used a convolutional neural network (CNN) based on a deep learning model that contains four convolutional and two fully connected (FC) layers.

CNNs first use a series of convolutional layers, each with multiple filters which extract features (e.g., lines, orientation, clustering) from small sections of the input image. Each layer has an increasing number of filters and finds higher levels of features (progressively incomprehensible to humans). Maximum pooling layers are added between convolutional layers to reduce the spatial size of the convolved feature, reducing the computational power required. This is achieved by passing a $2 \times 2$ pixel kernel over the image and extracting the maximum value, helping extract dominant features. After the convolution, the data are passed to multiple FC layers that learn which features are important in categorising the image. The final FC layer is the output layer which returns a categorisation of the input image along with a confidence in the result.

Figure 1 shows a simplified schematic of the CNN architecture we use to create our plume identification model. The input image is first passed through two convolutional layers with 32 and 64 filters, respectively, followed by a maximum pooling layer. We then randomly drop 50% of the data from the model, which helps to prevent over fitting of the data. The remaining data is then passed through two more convolutional layers of 128 and 256 filters, respectively, and another maximum pooling layer. This is followed by dropping another 50% of the data and flattening the array to one dimension to be fed into the FC layer that contains 512 nodes. Each CNN layer is then passed into a Rectified Linear Unit (ReLU) activation function before going into next layer. The last FC is passed into a softmax function to calculate the probabilities that the image contains a plume or not. The optimiser used here is an AdamOptimizer, which helps to reduce the cost calculated by cross entropy. The model has a total of 4,892,770 trainable parameters.



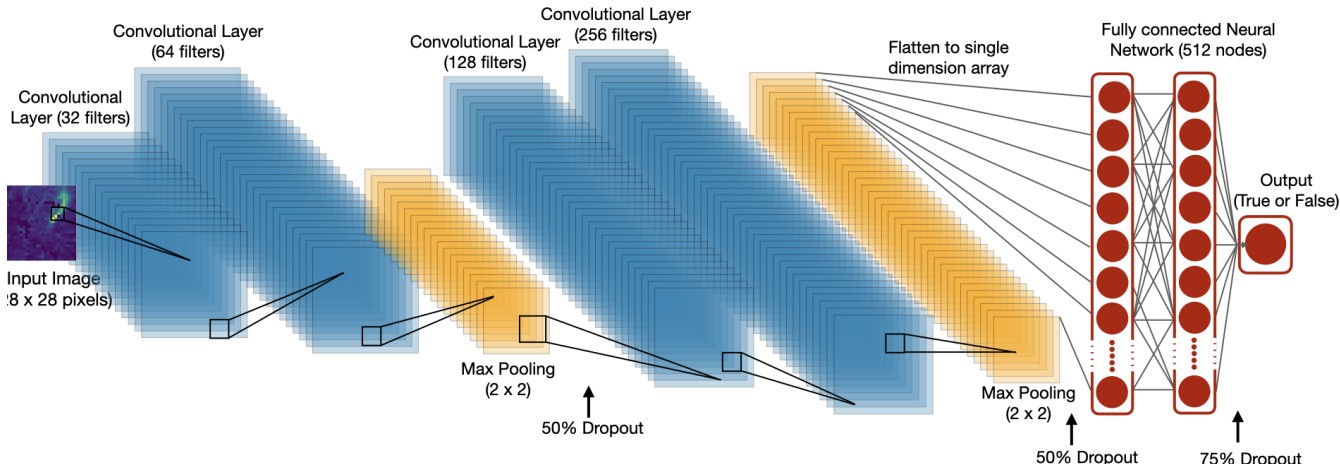

**Figure 1.** Simplified schematic of the convolutional neural network architecture to identify $NO_2$ plumes. See main text for further details.

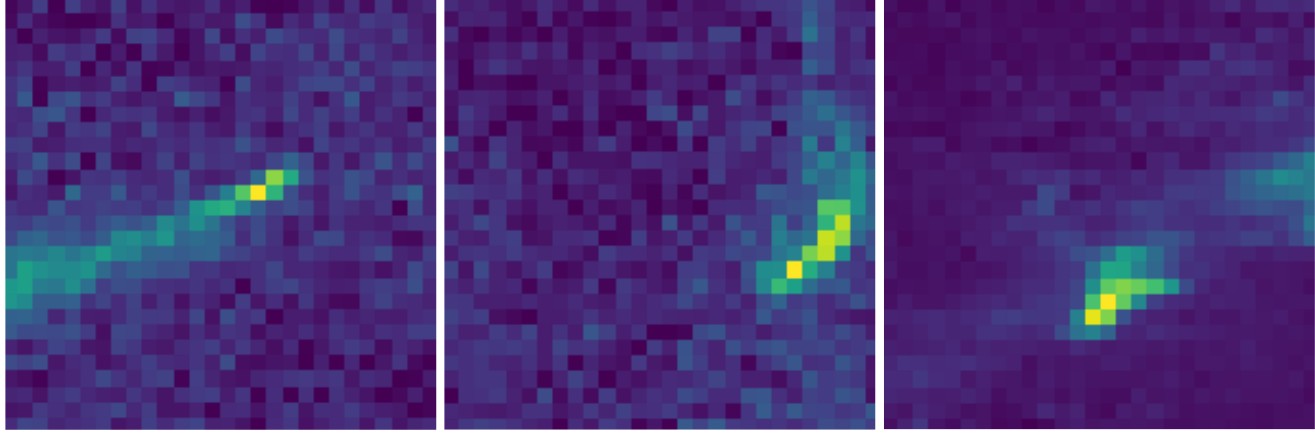

**Figure 2.** Three example individually normalized images of TROPOMI tropospheric column $NO_2$ that contain a plume.

**Supervised Learning Strategy**

To train our CNN model we use example images of TROPOMI $NO_2$ that can be classified as containing a plume or not. Each image is 28×28 pixels (approximately corresponding to 200 km×100 km) and were individually normalised to remove the influence the magnitude of image $NO_2$ features, a step that also improved model efficiency. We acknowledge that normalising each individual image could potentially lead to false detection if the background noise resembles a plume; the alternative of normalising the images to a standard value decreases the models ability to detect smaller emission sources and may lead to a larger number of false negatives. Figure 2 shows three example images from the level 2 TROPOMI $NO_2$ data considered to contain plumes used in the training dataset.





Determining whether an image contains a plume or not is a non-trivial task that is subject to human judgement and is consequently prone to error. Plumes are highly variable in both size and shape, can potentially be obscured by other features in the image, or, in some instances, multiple plumes can be found within a single image. In the first instance, we used a crowd-sourcing approach in which we posted images on https://plume-spotter.herokuapp.com and asked participants to determine whether an image contained a plume. A total of 41 participants classified 1565 unique images and created 13,750 classifications,

a mean of 8.8 classifications per image. This is further described in Appendix A. However, we found that this approach did not produce consistent results, with a larger number of images inconclusively classified by the participants than the number of images for which the participants agreed. This result emphasizes the role of human bias in identifying plumes, which in the absence of any post-training check compromises the performance of the CNN model.

     Due to the lack of agreement in our crowd-sourced approach to plume identification, we created a dataset for this study based

on the authors' judgement. We selected a total of 6086 images (3043 of which contained a plume) from across the globe for all times of year to minimize regional and seasonal biases. We used an iterative process to select images to train the model. We started with an initial set of images, randomly selected images that contained at least one plume, corrected the classification if necessary, ensuring an equal number of true and false images were included in the training set. The images were then randomly split in a 80:20 ratio to train the CNN model and test the trained model. We find that the resulting CNN model achieves an

accuracy of >90% when compared against the test data.

     Using the developed plume identification model, we processed two years (July 2018–June 2020) of TROPOMI tropospheric $NO_2$ data, resulting in 18 million individual $28 \times 28$ pixels images. Prior to running the model, we discarded images that included >40% invalid pixels, i.e., data that did not match our quality threshold as described above; this quality control step reduced the number of processed images to approximately 7.2 million. We then passed these images to our CNN model, which

returned a Boolean variable that describes whether a plume was identified and an associated confidence level associated with the identification. We discard images for which the confidence threshold <75%. We find that our results are moderately sensitive to this value, with an approximate 10% change in the number of plumes found when changing the confidence threshold by $\pm15\%$. For each image in which a plume was identified, we extract the geographical coordinates of the plume by identifying the image pixel with the maximum value. We acknowledge this method could lead to inaccuracies as the maximum pixel value in the image will not necessarily correspond to the origin of the plume, but we consider this as a minor source of error. The area

of one TROPOMI $NO_2$ pixel is approximately 24 km$^2$ so the plume origin could easily fall within this area. Each image has an associated timestamp from the satellite allowing us to build a dataset of the location and time of plumes spotted by TROPOMI.

## 3   Results

     First, we assess the performance of the CNN model to identify plumes on global and regional spatial scales. We then use the

locations of these plumes to study their ability to identify anthropogenic combustion sources of $CO_2$.



## 3.1 CNN Model Performance

Over our two year study period, the CNN model identified 310,020 images that contained at least one plume. After extracting the geographical locations for each plume location, we identified 62,040 (20%) images that were within 15 km of an active fire as determined by VIIRS thermal anomaly data and categorise these $NO_2$ plumes as being associated with biomass burning. We assign the remaining 247,980 $NO_2$ plumes as originating from anthropogenic combustion.

### Global Scale Plume Distributions

Figure 3 shows the location of $NO_2$ plumes from burning fossil fuels and biomass burning over our study period. We find anthropogenic combustion is widespread across the global (Figure 3a) with a focus over northern mid-latitudes, India and China, as expected. We also find coherent distributions of $NO_2$ plumes over the ocean along established shipping routes. On the global scale, this is particularly noticeable in the Bay of Bengal between southern India and Southeast Asia, and between the Cape of Good Hope and north-west Africa and eastern Brazil. Shipping lanes are clearer on the regional scales we report below.

The cluster of plumes over northern Alaska (Figure 3a) is an excellent example of a geographic region where $NO_2$ emissions are dwarfed compared to other point sources on a global scale so will not typically appear as a hot spot using other detection methods. We believe these are genuine detections, which we link to petroleum extraction activities in the National Petroleum Reserve–Alaska in the Alaska North Slope region.

The distribution of biomass burning $NO_2$ plumes (Figure 3b) identified using the CNN model and VIIRS data highlight geographical regions where we expect seasonal fire activity, with a high density of plumes over western, central and eastern Africa, Colombia, Venezuela, Brazil and over Australia.

We acknowledge that a number of plumes that we classify as anthropogenic combustion occur in locations where we expect biomass burning, e.g., central Australia, various regions across the tropics, Siberia, and North America. While this suggest that our use of VIIRS is imperfect we find that our approach broadly achieves its goal.

### Regional Scale Plume Distributions

Figure 4 shows anthropogenic combustion plumes we identify from July 2018 to June 2020 over Europe, contiguous US and southern Canada, China, and the Middle East. We have broadly classified these hotspots as major urban areas, power stations, and flaring regions. We identify major urban areas (populations >200,000) based on data from https://www.naturalearthdata.com/downloads/10m-cultural-vectors/10m-populated-places/, fossil fuel power stations taken from the Global Power Plant Database (KTH Royal Institute of Technology, 2018), and oil and gas flaring regions based on data from https://skytruth.org/flaring/. We acknowledge that the power station database will be incomplete due to data availability and reliability across the globe (Byers et al., 2019). The location of oil and gas flaring used here, determined by nighttime thermal anomaly data from VIIRS, is clustered spatially and temporally and therefore may not coincide with the TROPOMI local overpass time of 1330.



**Figure 3.** Geographical locations of individual TROPOMI NO$_2$ plumes identified using a CNN model, July 2018–June 2020. We attribute these plumes to a) anthropogenic combustion or b) biomass burning depending on whether the plume falls within 15 km of the nearest VIIRS thermal anomaly measurement.





We find the highest density of plumes are found over large cities, e.g., Paris, Madrid, Riyadh, Beijing, Los Angeles and New York, and over busy ports such as Rotterdam, Porto, Cairo and Hong Kong (4a, b, c). Ship tracks are clearly seen through the Strait of Gibraltar (Figure 4a) and the Red Sea leading to the Suez Canal (Figure 4d). Plumes over China, Korea, and Japan

are so dense they begin to overlap (figure 4c). We also find clusters of $NO_2$ plumes over and around power stations and flaring regions, with some notable exceptions, e.g., North Sea oil fields (Figure 4a), the oil fields in Oman and north-west Egypt (Figure 4d) and the large number of power stations in the Midwestern United States (Figure 4b). The poor correspondence between flaring regions and $NO_2$ plumes may be due to differences in the overpass times of the data used, as discussed above. In general, the location of the $NO_2$ plumes and the coincidence with cities, power plants, and established shipping routes provides us with

confidence of the CNN model we have developed. Discrepancies between known sources and the $NO_2$ plumes, especially over China and India suggest that inventories being used to identify power plants are out of date. Achieving this level of detail using conventional plume detection methods would be difficult.

Table 1 shows the top ten countries with the most fossil fuel plumes identified over the two year study period. China contains the most plumes, representing 20% of all the plumes found during our study period. These plumes are mainly located around

the highly urbanised and heavily industrial east of China (Figure 4c), encompassing Beijing, Hebei and Shenyang in the north east. India is a close second with 17% where most plumes are over New Delhi, Mundra Port in the north-western Gurjurat region, and large coal mining areas to the north-east of the country (Figure 3a). Russia is responsible for 12% of plumes, spread over multiple cities and fossil fuel extraction works across the west of the country. The Middle East, including Iran, Saudi Arabia, and Iraq, are collectively responsible for more than 26% of plumes. These plumes are mostly coincident with

known regions of petroleum extraction and processing. Values over eastern Egypt appear to follow the Nile and abruptly stop before Sudan. Plumes over the US mainly coincide with major urban areas and flaring regions, with clusters found over some of the major oil and gas extraction sites, e.g., San Juan Basin, Permian Basin, Niobrara Formation, and Bakken Formation. There is also some evidence of oil and gas extraction over Mexico, e.g. Burro Picachos and Sabinas, and over Kazakhstan, e.g. the Aktobe oil fields.

We acknowledge that statistics reported here will reflect the number of cloud-free days over specific regions. The frequency of global plume detections does change every month but does not show any seasonal cycle (not shown), even though there is a seasonal cycle of plumes detections at high latitudes due to low sun angles during winter. Over our study period, the monthly mean number of fossil fuel $NO_2$ plumes is 11,100 and biomass burning $NO_2$ plumes is 2,787. The largest number of fossil fuel plumes and biomass burning plumes were found during March 2019 and August 2018, respectively. Persistence of plume

detection locations (Figure 4) provide confidence that we observing point sources. We find a total of 21,802 plumes detected over the oceans, mostly focused along ships tracks.

Table 2 shows the 20 cities across the globe with the most fossil fuel plumes identified over our two year study period. As previously discussed, cities that are likely to have more high quality (cloud-free) retrievals are more likely to have plumes spotted over them, however the list of cities with the largest number of plumes is as expected based on knowledge of their large

emissions. All of these 20 cities are within latitudes 35°S–35° N. Six of the cities are in India with two each in neighbouring Pakistan and Bangladesh. Los Angeles and Phoenix are the only two US cities in the list, two in Mexico (Mexico City and

**Figure 4.** Geographical locations of individual anthropogenic combustion plumes (denoted by blue dots) identified from TROPOMI tropospheric NO$_2$ column data using a CNN model, July 2018–June 2020. a) Europe, b) North America, c) China, and d) Middle East. Also shown are the locations of major urban areas (populations >200,000) denoted by grey circles; coal, oil, and nature gas power stations denoted by red squares; and oil and gas flaring locations are shown as orange rectangles.



**Table 1.** Top ten countries containing the most fossil fuel plumes identified by TROPOMI $NO_2$ plumes, July 2018–June 2020.

| Rank | Country | Number of Plumes | % Total |
|------|---------|------------------|---------|
| 1 | China | 27290 | 19.9 |
| 2 | India | 23258 | 17.0 |
| 3 | Russia | 17225 | 12.6 |
| 4 | Iran | 16035 | 11.7 |
| 5 | Saudi Arabia | 14924 | 10.9 |
| 6 | USA | 13873 | 10.1 |
| 7 | Mexico | 7387 | 5.4 |
| 8 | Kazakhstan | 6197 | 4.5 |
| 9 | Egypt | 5389 | 4.0 |
| 10 | Iraq | 5336 | 3.9 |

Torreón), and two in South America (Buenos Aires and Santiago). The others in north Africa (Cairo, Egypt and Khartoum, Sudan). No cities in China are found in the top 20 despite most plumes being found in the country. We attribute this to large areas of industry in China being located outwith city boundaries.

## 3.2 What Fraction of Anthropogenic $CO_2$ Emissions are Identified Using $NO_2$ Plumes?

The low frequency of corresponding TROPOMI $NO_2$ measurements and satellite observations of $CO_2$ precludes any meaningful statistical analysis of of $CO_2$:$NO_2$ (not shown). We anticipate this improve with the launch of new satellites, particularly with the Copernicus $CO_2$ constellation (CO2M) due for launch in 2025 and the Japanese Global Observing SATellite for Greenhouse gases and Water cycle (GOSAT-GW) due for launch in 2023.

To help us understand the fraction of global anthropogenic $CO_2$ emissions that are identified using our plume identification model, we sample the Open-source Data Inventory for Anthropogenic $CO_2$ (ODIAC, 2020 release, (Oda and Maksyutov, 2011; Oda et al., 2018; Oda and Maksyutov, 2021)) where there is an $NO_2$ plume. We use the monthly $1° \times 1°$ ODIAC gridded land $CO_2$ emissions dataset for 2018 and 2019, and in the absence of 2020 data we use the 2019 ODIAC emissions for January–June 2020 to compare against the plume dataset. We sample the ODIAC dataset between -50°– 50° north to remove the impact of fewer observations during winter months. For this comparison, we assume that all anthropogenic combustion sources of $CO_2$ in the ODIAC dataset co-emit $NO_2$ and therefore can be used as geographical validation for the plume detection dataset.

We sample the ODIAC dataset at the location and month of each $NO_2$ plume identified using our model. Figure 5 shows the cumulative percentage of total emissions and the corresponding emission rate described as a function of the percentage of sources (from large to small) for all the ODIAC dataset and the ODIAC dataset sampled at the $NO_2$ plume locations. We find that for ODIAC emissions, large sources ($>1$ gC/m$^2$/day) account for approximately 25% of all sources but contribute





**Table 2.** Top twenty cities containing the most fossil fuel plumes over the study period.

| Rank | City | Number of Plumes |
|------|------|------------------|
| 1 | Delhi, India | 1015 |
| 2 | Los Angeles, USA | 738 |
| 3 | Dakha, Bangladesh | 726 |
| 4 | Cairo, Eygpt | 654 |
| 5 | Phoenix, USA | 537 |
| 6 | Lahore, Pakistan | 516 |
| 7 | Kabul, Afghanistan | 516 |
| 8 | Khartoum, Sudan | 491 |
| 9 | Surabaya, India | 462 |
| 10 | Rawalpindi, Pakistan | 412 |
| 11 | Kolkata, India | 409 |
| 12 | Ahmedabad, India | 378 |
| 13 | Chittagong, Bangladesh | 367 |
| 14 | Buenos Aires, Argentina | 360 |
| 14 | Santiago, Chile | 360 |
| 16 | Mexico City, Mexico | 354 |
| 17 | Torreón, Mexico | 349 |
| 18 | Surat, India | 325 |
| 19 | Mumbai, India | 321 |
| 20 | Seoul, South Korea | 321 |

approximately 90% to the total emissions. The ODIAC emissions sampled by the $NO_2$ plumes accounted for 92% of all global $CO_2$ emissions, described by 56% of all sources. The remaining 8% of emissions, described by 44% of all sources, typically have emissions <0.18 $gC/m^2$/day. This suggests that our method of identifying $NO_2$ plumes is biased towards the largest end of the emission spectrum and is less sensitive to the smallest emissions. This limit of detection does not lead to a large discrepancy

in the total emissions being sampled by the $NO_2$ plumes, reflecting the disproportionate role of large emission sources on the total emission budget.

Figure 6 shows the ODIAC emissions where no $NO_2$ plumes where detected. Out of these undetected sources, 95% have emission rates <0.18 $gC/m^2$/day and only nine locations have an emission rate >1 $gC/m^2$/day denoted by the green circles. Five of these locations are situated in the USA, with the remaining in Colombia, China, Japan and Slovenia. The locations

in the USA are all between large cities, connected by highways that are not described by single point sources. The source

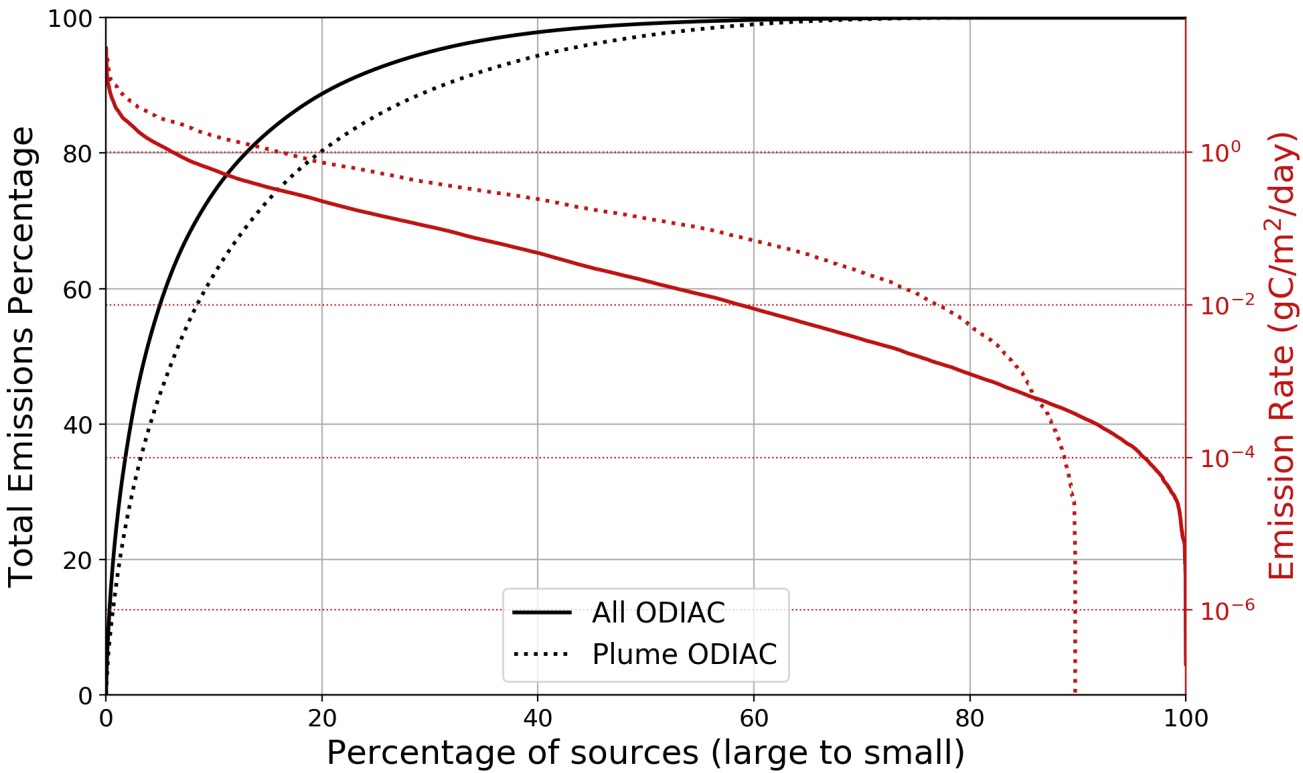

**Figure 5.** The cumulative total emission percentage as function of source size (black) and emission rate as a function of source size (red) for all ODIAC (solid line) and ODIAC sampled at plume locations (dotted line).

in Colombia is located in an area of persistent cloud cover (>80% of the year) and therefore will have fewer high quality observations from TROPOMI. The reason for the missed large sources over China, Japan and Slovenia are unclear.

Figure 5 also shows that 10% of our plumes do not correspond to ODIAC $CO_2$ emissions. This is due mostly to plumes over the ocean associated with ship tracks (Figures 3 and 4), but there will be instances where fires have not been removed using
our VIIRS criterion (described above) and possibly false detections. Here, we also consider the possibility that the emission inventory is incomplete for some reason.

Figure 7 shows four examples of clusters of plumes spotted by the detection method which do no have any associated ODIAC emissions. The clustering of the plumes suggest they are highly unlikely to be false detections and the persistence of the features over multiple years suggests it is unlikely to be biomass burning. Although accurate determination of the emissions associated
with these hotspots is outside the scope of this paper, we hypothesize based on satellite imagery from Google Maps that these are regions of fossil fuel extraction and processing (coal in China and oil and gas in Mali, Saudi Arabia and Iraq). Having the ability to detect these plumes automatically provide a method of frequently updating emission inventories. Although these



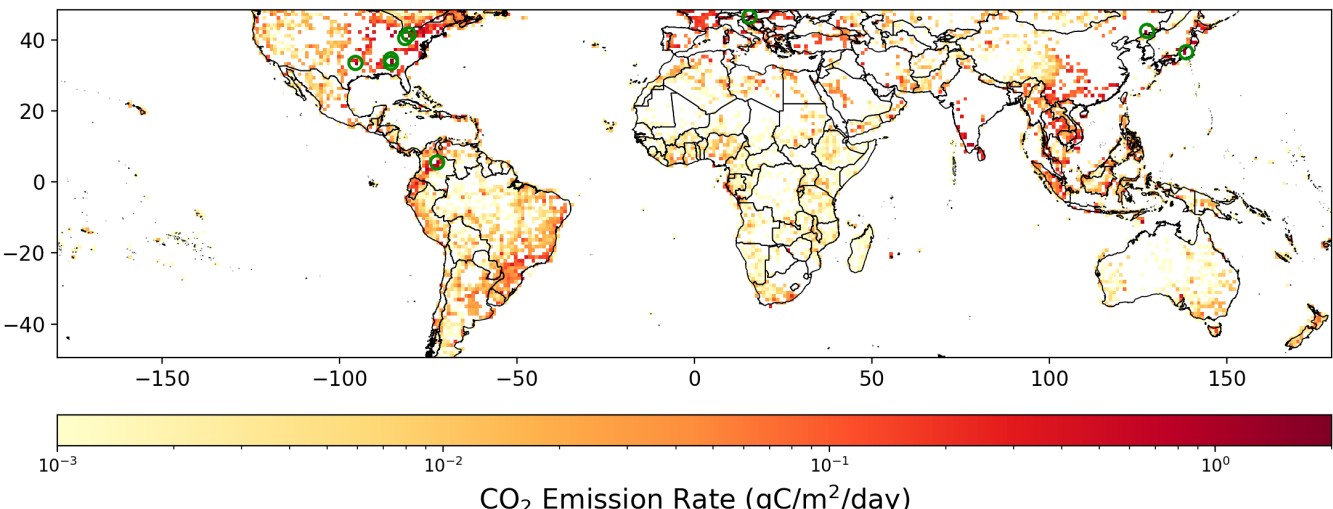

**Figure 6.** $CO_2$ emission rates as per the ODIAC inventory where no plumes were detected. The green circles indicate sources greater than 1 $gC/m^2/day$.

clusters of plumes could be persistent errors from highly reflective features such as salt lakes and solar panels, it is unlikely that they would appear as plume shaped anomalies and therefore are less likely to be picked up by the CNN model.

## 4    Discussion and Conclusions

We have developed a convolutional neural network (CNN) to identify plumes of atmospheric nitrogen dioxide ($NO_2$), a tracer of incomplete combustion. We have trained the model using a small subset of available images from the TROPOspheric Monitoring Instrument, aboard Sentinel-5P. The resulting CNN reveals a rich distribution of plumes across the globe, which correspond to large city centres, power plants, oil and gas production, and shipping routes. Many of these features would be difficult to isolate without the use of a deep learning model.

The impetus for our study is using $NO_2$ as a tracer for anthropogenic emissions of $CO_2$ and methane. The main advantage of using $NO_2$ is its comparatively short atmospheric lifetime, allowing to relate elevated values to local emissions. We have attempted to remove biomass burning using thermal anomaly data, which is often used to locate open biomass burning. This is not a perfect method, but our results suggest it works reasonably well. To evaluate our ability to observe anthropogenic emissions of $CO_2$ we have used the Open-Data Inventory for Anthropogenic Carbon dioxide (ODIAC) (Oda et al., 2018), an established emission inventory used widely by the community. We have chosen this approach because we found the number of coincident measurements of TROPOMI $NO_2$ and OCO-2/GOSAT $CO_2$ was not sufficient to generate meaningful statistics (not shown). By sampling ODIAC at the location of $NO_2$ plumes, we find that the CNN model describes 92% of global anthropogenic $CO_2$ emissions. The remaining 8% of emissions, mostly $<0.2$ $gC/m^2/day$, provide an effective limit of detection for our method.



**Figure 7.** Example locations with plume clusters that are not associated with ODIAC $CO_2$ emissions. The light grey lines show major roads and urban areas are shown by grey patches.



Our use of $NO_2$ to describe anthropogenic emissions of $CO_2$ and methane relies on them being co-emitted. We find no evidence in the literature of $NO_x$ scrubbers being used for power plants, although they are used by the chemical industry, which is a sector that represents a comparatively small emission of $CO_2$. The validity of using $NO_2$ as a proxy for $CO_2$ emissions may change in the future as non-catalytic reduction and low-$NO_x$ burner technologies begin to mature. We find no

correlation between $NO_2$ plumes and the location of natural gas flaring, which is unexpected since this will be an inefficient form or combustion and therefore should result in a significant source of $NO_2$. We have no explanation for this observation, except if flaring occurs at preferential times of day that do not coincide with the early afternoon overpass time of TROPOMI. Our approach will also miss direct $CO_2$ and methane emissions, e.g., pipeline leaks, coal mines (Palmer et al., 2021). For these sources, we still have to rely on high spatially-resolved $CO_2$ and methane data (Varon et al., 2021). In contrast, we also

find persistent $NO_2$ plumes from regions where ODIAC does not currently include $CO_2$ emissions. Based on the location and inspection of satellite imagery provided by Google Maps we suggest these are likely to be associated with new areas where fossils fuels are being extracted or combusted for energy generation. This demonstrates how $NO_2$ plumes could be used to inform emission inventories about the location of new point sources across the globe.

The $NO_2$ plume detection algorithm does not quantify anthropogenic emissions of $CO_2$ or methane, but it provides a method

to refine the development of measurement, reporting and verification systems that form the backbone of the Paris Agreement. The launch of Copernicus $CO_2$ service, including a constellation of satellites that will measure $CO_2$, methane, and $NO_2$, will result in a step-change in the number of coincident measurements and thereby will improve our ability to use simultaneously use $NO_2$ with $CO_2$ and methane to quantify anthropogenic emissions of $CO_2$ and methane.

*Data availability.* TROPOMI $NO_2$ data is available from https://scihub.copernicus.eu, VIIRS biomass burning data is availble from https:

//firms.modaps.eosdis.nasa.gov/download/ and the ODIAC emission dataset is availble from http://www.odiac.org/index.html.

## Appendix A: Supervised Learning Using Crowd Sourcing

We created an online tool (https://plume-spotter.herokuapp.com) that briefly described what a plume is with a few examples of what they can look like and then displays 18 images in a 6×3 grid. These images were selected at random from an initial 1565 unique images which were compiled by the authors. Each participant was then invited to click on the images in the grid

which they considered to contain a plume and then to submit their selection. Once their results were submitted, the participant was asked to classify 18 more random images. We then used these results to determine how many true (contains a plume) or false (does not contain a plume) classifications each image received. We designed this method to reduce the amount of human error and individual judgement on what could be considered a plume or not.

A total of 41 participants classified 1565 unique images and created 13,750 classifications, a mean of 8.8 classifications per

image. The number of classifications per participant is unknown as this was an online tool open to the public and relied on how much time they were willing to give.



For this crowd sourcing experiment, there were approximately 580 images for which 0–10% of classifications were true, i.e. high confidence that these images do not contain a plume. There were approximately 130 images for which 90–100% of classifications were true. For the remaining (≃800) images there was little agreement between the participants about whether

they included a plume or not. Since the majority of the images from the initial dataset did not have a high level of agreement on whether they contained a plume or not, we decided that this dataset was not unsuitable to train our model.

Going forward, this experiment could be refined to help improve the results and give us more confidence in the classifications of the images. The experiment assumed all images did not contain a plume unless the participant changed the classification, this meant that if a participant did not see an image then it would be considered not to contain a plume. We also noticed that

what was considered a plume changed depending on the surrounding images. If the participant is unsure whether an image contains a plume or not, they may be more likely to keep the image classification as false if a surrounding image contained a clearer plume, or vice versa for is the surrounding images definitely did not contain a plume.

*Author contributions.* DPF and PIP designed the research; DPF prepared the calculations with initial input from TZ; DPF and PIP analyzed the results and wrote the paper, with comments from TZ.

*Competing interests.* No competing interests are present.

*Acknowledgements.* DPF, PIP and TZ gratefully acknowledge funding from the National Centre for Earth Observation funded by the National Environment Research Council (NE/R016518/1).



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
