# Peer review of "Automated detection of atmospheric NO2 plumes from satellite data: a tool to help infer anthropogenic combustion emissions"

_Atmospheric Measurement Techniques, 2021_

## Author Response (AR1)

We thank the two reviewers for providing useful comments on the manuscript, which has improved the clarity of the text. Below, we address all individual reviewer comments. Original reviewer comments are in italics.

**Reviewer 1**

*## General comments*

*The manuscript entitled 'Automated detection of atmospheric NO2 plumes from satellite data: a tool to help infer anthropogenic combustion emissions' by Finch et al. presents a convolutional neural network to identify plumes of NO2 from TROPOMI. The authors claim that the algorithm can be used for detecting individual plumes (urban, oil and gas production, power plants) and the distribution of the detected plumes was compared with an anthropogenic CO2 emission inventory. I found the approach is appealing, especially with regard to combining VIIRS data to sort out locations of open biomass burning. Furthermore, the attempt to correlate NO2 plumes with anthropogenic CO2 emission can be an interest of many readers in AMT. Notwithstanding the possible global application of such algorithms, I would suggest drawing conclusions more carefully by stating possible false detections of plumes caused by either the proposed model itself or the TROPOMI retrieval algorithm.*

This is a good point. Since receiving this comment from the initial review, we have increased the text dedicated to false positive detection. We have now added this caveat to the concluding remarks to ensure the reader appreciates this point.

*## Specific comments*

*Line 89. I am not sure whether this 'active fire' (VNP14 data) can be identical to 'open biomass (or fossil fuel) burning. Maybe further explanations or rationales may be useful.*

The reviewer is correct in stating that the active fire product is not identical to open biomass or fossil fuel burning. We use this product as a proxy for fire and have amended the text to clarify this point.

*Line 114. Can you please elaborate regarding the random drop of 50% of the intermediate features (not the data)? As far as I understand, this random dropout is not learnable. This means that you could actually reduce the number of convolutional layers before dropping the features randomly.*

The reviewer is correct that this should be "randomly drop 50% of the features (or layers)" not the data. This has now been corrected in the text.

Randomly dropping a subsample of the features is a common and simple way of preventing overfitting of the model. This has the effect of thinning the network which in turns requires the subsequent layer to apply more or less weight to the inputs of that layer. This forces each node in the layer to specialise, helping the model to become more general and prevents nodes from

co-adapting which can lead to overfitting. Although reducing the number of layers would also thin the network, this would give less opportunity for the model to learn features in the image. We have now included a reference to this paper (https://jmlr.org/papers/volume15/srivastava14a/srivastava14a.pdf) in the manuscript if readers wish to know more about the drop out stage. There are a huge number of different configurations possible when creating the model and we came to this configuration through multiple iterations and changes. It may be possible to refine the model further in both accuracy and efficiency, however this would be a task for future work.

*Line 122. What does this mean 'individually normalized'? Does it mean it was normalised per image? If it is true, doesn't it increase the possibility for false detections? What happens if you normalise TROPOMI data globally? What is the benefit of normalizing per image?*

The data was normalized per image. This has the potential to increase false detections if the background noise resembles a plume shape. However, the alternative is to normalize the data relative to the highest possible value in all the data and then the model will struggle to detect small sources, increasing the likelihood of false negatives. This point has now been made clearer in the text.

*Line 125-140. A machine learning algorithm is basically 'training data' itself. It seems that the training data were selected by 'crowd sourcing', and then by authors. Is it correct? If yes, why is that? Why not using actual distribution of plumes (or several known plumes)? Please discuss this point.*

The dataset used to train the model was not crowd sourced. We tried to gather a dataset via crowdsourcing and found that it introduced a lot of uncertainty into what was considered a plume (which would then be passed into the model). In the end, we developed our own training set for these data as we are unaware of any dataset that matches the requirements of 1) having enough data of the correct pixel dimensions across the globe with an equal number of plumes/no plumes in the image, and 2) an assortment of different shapes and orientations.

*Line 232-234. '2019 ODIAC emissions were used for January-June 2020' – How about the effect of COVID lockdowns during 2020? Can you also mention about this?*

We have added the following to the manuscript: "We do not anticipate that the COVID-19 related lockdowns of 2020 will significantly impact our results as the reduction in CO$_2$ emissions were less than expected \cite[]{Tollefson2021}"

*Line 189-290, 287-290, and Figure 7. I would suggest to examine carefully these detected clusters with other data sources (even Google maps). Couldn't these be possible errors from TROPOMI retrieval algorithms? For instance, reflection from salt lakes, solar panels,,, etc.. ?*

We examined these data sources using Google Earth Images as mentioned on line 275, and present our hypothesis about what they could be although the satellite imagery on Google Maps can be several years out of date. Reflections from features such as salt lakes and solar panels

may have made it through the TROPOMI quality control process, however it is unlikely that these features would show up as plume shaped anomalies and therefore would be less likely to be picked up by the model. Further examination with more up to date sources (e.g. Sentinel-2 imagery) is outside the scope of this paper but this demonstrates the potential of this method to locate new sources. This has also now been emphasised in the modified text.

**technical corrections**

*Line 83. 'the the Copernicus'*

This has now been corrected.

*Line 85. the spatial resolution of TROPOMI has been changed since 6 August 2019 (https://sentinels.copernicus.eu/web/sentinel/data-products/-/asset_publisher/fp37fc19FN8F/content/sentinel-5-precursor-level-2-nitrogen-dioxide)*

Although the spatial resolution of TROPOMI was increased to 3.5 x 5.5 km as of the 6th of August 2019, the original resolution is still available to use for all dates. We decided to keep with this for consistency across the study period as the model may need retraining on different resolution data. This is something to consider for future developments. We have now included this information in the manuscript.

*Line 106. Is '(progressively incomprehensible to human)' part necessary?*

We believe this informs the reader that the features found by the model are not necessarily obvious or comprehensible and therefore makes connections which go beyond what a human might be able to do.

*Line 186. 'overpass time of 1330' to 'overpass time of 13:30'*

This has been changed

*Line 301. 'and then and then displays'*

This has now been corrected.

**Reviewer 2**

*General comments*

*The paper of Finch et al. entitled 'Automated detection of atmospheric NO2 plumes from satellite data: a tool to help infer anthropogenic combustion emissions' examines the potential of using a deep learning method to detect plumes in satellite NO2 retrievals. This paper is a nice piece of work with a novel approach. Although their work on the plume detection is very solid I do have some critical remarks on the relationship with CO2 emissions. Nevertheless, the*

*developed method seems promising and with more satellite instrument coming into place this manuscript is very relevant for the scientific community.*

*Specific comments*

*In the introduction the authors describe the importance of establishing a national CO2 emission baseline as starting point for climate mitigation efforts. Although I agree this baseline is very important, I would like to point out that the reported annual country-level emissions of fossil fuel CO2 are very accurate. However, when looking at specific (urban) areas or facilities and/or at shorter time scales the uncertainties increase. As in the remainder of the manuscript the focus is on plumes from urban centres and industrial facilities I would stress this difference to clearly describe the importance of this work. Also on pg. 2, lines 32-33 'Compiled inventories, which rely on self-reporting, provide estimates on these emissions but rely on assumptions that can sometimes lead to inaccurate values.'*

We have added the following to the introduction:

"The importance of accurate emission estimates becomes even more prevalent at smaller geographical and temporal scales. Reported annual country level emissions of $CO_2$ tend to be reasonably accurate, but are typically not sufficiently detailed to support targeted policy development. Given the importance of establishing accurate national and sub-national emission baselines from which to reduce emissions as part of the PA, it is essential we have a robust measurement-based approach to estimate emissions of $CO_2$ and methane to complement inventory estimates."

*At several places in the manuscript the authors say that NO2 is a tracer of incomplete combustion, but strictly speaking this is not true and I would rather say a tracer of fossil fuel combustion. The authors explain this well on pg. 2, lines 51-53. Also in the discussion on why plumes from natural gas flaring are lacking this is mentioned and I doubt whether this conclusion is valid.*

This is a good point. We have changed the text to now read "a tracer of combustion" rather than "a tracer for incomplete combustion" as NO2 is also emitted during complete combustion. We have also amended the discussion so it does not say that the reason flaring produces large amounts of NO2 is due to it being an inefficient form of combustion. It now reads: "We find no correlation between $NO_2$ plumes and the location of natural gas flaring, which is unexpected since this will be an major form of combustion and therefore should result in a significant source of $NO_2$"

*Pg. 5, lines 153-154: The authors describe that the plume coordinate is determined by looking for the maximum value. What does this mean for images which contain multiple plumes? Also, I'm wondering whether a difference in performance exist for images with one vs. multiple plumes. Could the authors also indicate how many of these images contain multiple plumes?*

Our method cannot currently differentiate between images that contain one or more plumes. However, our method of defining the plume centre that uses the maximum value would lead to

plumes not being identified in images where there are multiple plumes. This is something we will address in further work and requires substantial model development which is outside the scope of this paper. We have amended the following sentence: "We acknowledge this method could lead to inaccuracies as the maximum pixel value in the image will not necessarily correspond to the origin of the plume and may not identify all plumes where images contain multiple plumes, but we consider this as a minor source of error."

*Pg. 7, lines 180-181: I think it's also likely that the reverse is true, namely that anthropogenic emissions are incorrectly discarded as biomass burning emissions, while they can easily be co-located. This leads me to the question what the goal of this exercise exactly is. Do the authors aim to detect plumes that are almost certainly anthropogenic and use that for verification of those specific locations? Or is the goal to detect as many anthropogenic plumes as possible for a full verification of global or national emissions? This is also related to their decision to remove images with a <75% confidence that a plume is present in that image. Could the authors reflect on this?*

Agreed. We already acknowledge that anthropogenic plumes may have been incorrectly labelled as biomass burning emissions. As stated in the text, our method of differentiating between biomass burning and anthropogenic plumes is imperfect. This problem will especially be a problem where both emission types are co-located. We have adjusted the text to make this clearer.

The goal of this study is to find as many plumes as possible, as accurately as possible, which represents a trade-off. Once this has been achieved (or close to it), we show a method that can help distinguish anthropogenic emissions which can be used to locate, and potentially verify sources across the globe. We envisage that most steps in our methodology will be refined over time and subsequent studies. There are numerous options of how to use these data. There is potential to collate the information on a national and global scale to verify emissions but also to find and verify local emissions at any location (satellite permitting).

The <75% confidence limit is also something that can be refined at a later date or adjusted depending on the way the data is being used for a particular study. We made the decision to use 75% as it generated a large enough dataset that we could demonstrate the usefulness of the tool but not introduce too many errors. We have added the following text to explain this: "This confidence limit can be adjusted to change the ratio of number of plumes spotted to the confidence in the results."

*Pg. 9, lines 200-201: 'Discrepancies between known sources and the NO2 plumes, especially over China and India suggest that inventories being used to identify power plants are out of date.' This could be one explanation, but given the authors' conclusion that 92% of the CO2 emissions are covered with their methodology it also seems likely that the missing sources are rather small and therefore more difficult to detect.*

We have amended the text to make it clearer that we refer to areas where there is a large number of plumes - reducing the chance of it being due to a small source. We have also

amended the text to say that power plants, identified by the inventory, that do not have any associated plumes could also suggest the inventory is out of date and that further discrepancies maybe due to sources outwith the inventories used in this analysis (e.g. small settlements with large industrial emissions).

*Pg. 9, lines 219-220: 'Persistence of plume detection locations (Figure 4) provide confidence that we observing point sources.' Could the authors indicate how often the same location is sampled on average and is there a seasonality in the detection of certain sources? Later in the manuscript the authors compare the detected sources and total emissions with monthly CO2 emissions and therefore the timing may play a role. I also wonder whether the 92% of CO2 emissions covered by the plumes are based on annual emissions?*

Locations will be sampled daily as the satellite passes overhead (occasionally more often when swaths overlap). As mentioned earlier in the paragraph, these observations will be subject to quality control and cloud cover so cloudier locations will have an accurate observation less frequently. There is also a seasonality at high latitudes where sunlight becomes an issue during winter months.

It is unlikely that a single source will have a plume detected every day as there might not be an adequate observation or the shape of the plume may not be obvious and not captured by the model. An exploration into what can be considered the same plume occurring repeatedly and what can be considered as "persistent" would be of interest in future work for determining what sources are producing the plumes.

The only marked seasonality observed in our detection of plumes was due to biomass burning. This is to be expected and for our study was driven by the seasonality in the VIIRS dataset that was used to determine which plumes are biomass burning. We found no seasonality in anthropogenic plumes.

We report 92% of the ODIAC emissions covered by the plumes on a monthly basis, i.e. when a plume is detected, are there ODIAC emissions in that location that month? We believe this will account for any seasonality on the ODIAC dataset.

*Pg. 14, line 276: 'The impetus for our study is using NO2 as a tracer for anthropogenic emissions of CO2 and methane.' Methane has not been mentioned before in the manuscript (except for the introduction) and I would like to point out that the conclusions drawn here for CO2 may not apply to methane. The emission sources of methane are very different and therefore also the relationship between NO2 plumes and CH4 emissions. More nuance is needed in this statement.*

Agreed. Methane emissions would have a very different relationship with NO2. We have amended the text to emphasise that this work would provide a basis to explore further pathways in developing emission estimates of CO2 and methane from combustion.

*Pg. 16, lines 299-300: I agree with this statement, but I would rather move this to the introduction. Now the introduction seems to suggest that the authors want to establish a CO2 baseline emission, which is in fact not true.*

We have now included a version of that sentence to the introduction: " Although the NO2 plume detection algorithm does not quantify anthropogenic emissions of CO2 or methane, it provides a method to refine the development of future MVR systems which can directly feed into policy decisions."

Technical corrections

*Pg. 2, line 39: Please update the reference in this line (International Marine Organization).*

This has been corrected.

*Pg. 5, lines 124-125: '… were individually normalised to remove the influence the magnitude of image NO2 features, …' Please correct this sentence.*

This has been corrected to " ...to remove the influence of the magnitude of NO2 features…"

*Pg. 7, line 167: '… from burning fossil fuels and biomass burning…' Please replace with 'burning of fossil fuels' or 'fossil fuel and biomass burning'.*

This has been changed to "fossil fuel and biomass burning"

*Pg. 7, line 168: '… across the global…' Replace 'global' with 'globe'.*

This has been changed

*Pg. 9, line 220: '… that we observing…' Please correct this sentence.*

Corrected to "that we are observing…"

*Pg. 11, line 232: 'We anticipate this improve…' Please correct this sentence.*

Corrected to "We anticipate this will improve…"

*Pg. 16, lines 290-291: '… an inefficient form or combustion…' Please correct this sentence.*

Corrected to "an inefficient form of combustion…"

---

## Author Response (AR2)

**Automated detection of atmospheric NO$_2$ plumes from satellite data: a tool to help infer anthropogenic combustion emissions**
**AMT - Reviewer Response**

*Major comments:*
*Line 40. It would be helpful to clarify how and why higher resolution emission estimates in time and space are necessary.*

We have now clarified this statement:

*The importance of accurate emission estimates becomes even more prevalent at smaller geographical and temporal scales for which, for example, hotspots and diurnal variations will play a larger role in driving observed atmospheric concentrations.*

*Line 132 and Figure 2, the example plum images presented here seem to be subject to winds. How does a plum looks like when there are no winds? downwind, upwind etc.. Are these considered when the authors created the training data ('the ground truth' )? How the meteorological conditions and 'the ground truth' introduced to the model are related? Would the authors like to claim that such conditions related to meteorology would not be a problem when averaging across the study period (time-averaging for detecting plum locations only)?*

This is a good question. We have clarified this point in the text after Figure 2:

*For an emission source to create a plume detectable by TROPOMI, the source must be subject to winds strong enough to disperse the emissions across multiple pixels within the lifetime of NO2. We anticipate that the number of occurrences where these conditions are not met will be relatively small compared to the entire dataset and therefore should not have an adverse effect on our results.*

*Line 144. In addition to the point above, a discussion regarding the authors' judgement should be presented in the manuscript.*

We have added the following statement:

*Subjective judgement of the images could lead to small variations in repeated experiments and therefore a more rigorous approach may be needed for future applications.*

*Line 184. If the seasonal dependency is expected but not presented, can you elaborate a bit more about the temporal aspect of the presented approach?*

Although we present all detections over the two year study period, we attribute plumes to biomass burning based on daily VIIRS biomass burning data - corresponding to the date of the plume. This means seasonality of the location of the biomass burning plumes will be present in the final dataset. We have added the following statement to clarify:

*We account for the seasonal variation of fire activity by using daily VIIRS data, and remove fire-influenced scenes from those identified by our CNN.*

*Line 278-281. I appreciate the authors included the raised concern from the last review. However, it would be helpful to mention general issues for false detections, which can be caused by TROPOMI retrieval or the model (authors' judgement) itself*

We have included the following statement:

*As well as errors in the TROPOMI retrieval leading to false detections in the final dataset, errors may also occur during the creation of the model (e.g. mislabelled training data). A single plume data point may not represent a real-life plume and should be considered in context of other data (e.g. frequent recurrence, land use, proximity to other sources). Further refinement of the training dataset, model parameters and data analysis stages will reduce the number of false detections and feedback to the TROPOMI community could help reduce the number of retrieval errors.*

*Minor comments:*
*Line 33. The authors mentioned that the emission inventories rely on assumptions that lead to in-accurate values. Which assumptions can lead to inaccuracy?*

We have changed the sentence to read:

*Compiled inventories, which rely on self-reporting, provide estimates on these emissions but rely on assumptions such as fuel consumption, combustion efficiencies and emission rates that will be imperfect.*

*Line 36-37. The authors claim that China and India are going to build new coal-fired power plants. Please add references.*

Given the timelines associated with the peer-review process, these references are in the "reputable" mainstream press and grey literature. For example,
https://time.com/6090732/china-coal-power-plants-emissions/
https://www.theguardian.com/environment/2021/oct/12/china-coal-fired-plants-uk-cop26-climate-summit-global-phase-out
https://carbontracker.org/reports/do-not-revive-coal/
https://www.reuters.com/business/energy/cop26-aims-banish-coal-asia-is-building-hundreds-power-plants-burn-it-2021-10-29/

With this in mind, we think it is appropriate not to include any one reference to accompany our statement.

*Line 92. The authors mentioned that regardless of changes in TROPOMI resolution from August 2019, the original resolution was used. Is it possible to use the same resolution when using*

*Level2 data? Or Does it mean that you re-gridded the TROPOMI data with the same resolution to make each pixel the same distance?*

This statement contained an error in that the 7x3.5 km resolution is the higher resolution product that became available from August 2019 but the updated processing was retroactively applied to the data collected before that date so all data used is of the higher resolution.

We have removed the line to avoid confusion:
'*Higher resolution data are available from August 2019, but for consistency we have used the original resolution throughout our study period.*'

*Line 120. references missing*

There does not appear to be a reference missing on this line. We reference Srivastava et al. (2014) in regards to dropping convolutional layers in the model.

*Line 121. dropping another 50% of the 'layers'*

Changed.

*Line 129. For the normalization of images, the authors mentioned it was to improve model efficiency. Is it correct? Isn't it a 'necessary' step for training the model without diverging?*

We have updated this line to read:

*Images...were individually normalised to remove the influence of the magnitude of NO2 features, a step that also ensures the model parameters have a similar data distribution and therefore improves the model efficiency and accuracy.*

*Line 152. 'our quality threshold' -- does it mean the TROPOMI quality flag?*

We do mean the TROPOMI threshold and the text has been amended.

*Line 242. 'analysis of of'*

Changed.

*Line 250. references missing*

As this is our assumption based on the work referenced in the introduction we do not think a reference is needed here.